# Toxicity of Cyanopeptides from Two *Microcystis* Strains on Larval Development of *Astyanax altiparanae*

**DOI:** 10.3390/toxins11040220

**Published:** 2019-04-13

**Authors:** Kelly Fernandes, Andreia Gomes, Leonardo Calado, George Yasui, Diego Assis, Theodore Henry, Ana Fonseca, Ernani Pinto

**Affiliations:** 1Natural Resources Institute, Federal University of Itajubá, 1303 BPS Avenue, Itajubá, MG 37500-903, Brazil; kelly.af@usp.br (K.F.); andreia.gomes@ifrj.edu.br (A.G.); afonseca@unifei.edu.br (A.F.); 2School of Pharmaceutical Sciences, University of Sao Paulo, 580 Professor Lineu Prestes Avenue, São Paulo, SP 05508-000, Brazil; 3Federal Institute of Education Science and Technology of Rio de Janeiro, Washington Luis Highway, Niteroi, RJ 24310-000, Brazil; 4National Center for Research and Conservation of Continentals’ Fish—CEPTA, SP-201 (Pref. Euberto Nemésio Pereira de Godoy—Motorway), Km 6.5, Pirassununga, SP 13630-970, Brazil; leonardocalado@hotmail.com (L.C.); yasui@usp.br (G.Y.); 5Faculty of Technology, State University of Campinas, 1888 Paschoal Marmo Street, Limeira, SP 13484-332, Brazil; 6Bruker Daltonics Corporation, Condomínio BBP—Barão de Mauá, Atibaia, SP 12954-260, Brazil; diego.assis@bruker.com; 7Institute of Life and Earth Sciences (ILES), Center for Marine Biodiversity & Biotechnology (CMBB), The School of Energy, Geoscience, Infrastructure and Society (EGIS), Heriot-Watt University, Edinburgh EH14 4AS, UK; t.henry@hw.ac.uk

**Keywords:** cyanobacterial peptides, mass spectrometry, microcystins, *Astyanax*, morphological alteration, GNPS

## Abstract

Absorption and accumulation of bioavailable cyanobacterial metabolites (including cyanotoxins) are likely in fish after senescence and the rupturing of cells during bloom episodes. We determined the toxicity of cyanopeptides identified from two strains of *Microcystis* (*M. panniformis* MIRS-04 and *M. aeruginosa* NPDC-01) in a freshwater tropical fish, *Astyanax altiparanae* (yellowtail tetra, lambari). Aqueous extracts of both *Microcystis* strains were prepared in order to simulate realistic fish exposure to these substances in a freshwater environment. Both strains were selected because previous assays evidenced the presence of microcystins (MCs) in MIRS-04 and lack of cyanotoxins in NPDC-01. Identification of cyanobacterial secondary metabolites was performed by LC-HR-QTOF-MS and quantification of the MC-LR was carried out by LC-QqQ-MS/MS. MIRS-04 produces the MCs MC-LR, MC-LY and MC-HilR as well as micropeptins B, 973, 959 and k139. NPCD-01 biosynthetizes microginins FR1, FR2/FR4 and SD-755, but does not produce MCs. Larval fish survival and changes in morphology were assessed for 96 h exposure to aqueous extracts of both strains at environmentally relevant concentrations from 0.1 to 0.5 mg (dry weight)/mL, corresponding to 0.15 to 0.74 μg/mL of MC-LR (considering dried amounts of MIRS-04 for comparison). Fish mortality increased with concentration and time of exposure for both strains of *Microcystis*. The frequencies of morphological abnormalities increased with concentration in both strains, and included abdominal and pericardial oedema, and spinal curvature. Results demonstrate that toxicity was not solely caused by MCs, other classes of cyanobacterial secondary metabolites contributed to the observed toxicity.

## 1. Introduction

The occurrence of cyanobacterial blooms and the presence of some water soluble-cyanotoxins are well documented in several inland water bodies worldwide [1]. Cyanotoxins such as cyclic peptide microcystins and nodularins, lipopolysaccharides and some alkaloids (i.e., anatoxins, saxitoxins, cylindrospermopsins and lyngbyatoxins) are well known and produced by common freshwater cyanobacterial species (i.e., *Microcystis*, *Cylindrospermosis* and *Anabaena* genera) [2]. Almost all classes of cyanotoxins have been studied including their main mechanisms of action, and environmental risk assessments are typically focused on these known cyanotoxins [3].

Although cyanotoxins are important, cyanobacteria also produce a wide range of biologically active secondary metabolites for which only some classes have been fully characterized and assessed for toxicity [4]. Extracts and natural products isolated from freshwater phytoplankton species, including eukaryotes and cyanobacteria, have been screened in vitro for specific biological action, including UV protection and inhibition of particular classes of proteases [5,6,7]. There are relatively few in vivo studies on the effects of secondary metabolites of cyanobacteria, but those that have been conducted indicate these substances can be important, including for the detection of allelopathy in some cyanometabolites with other competitive species in the environment, antifeeding and antibiotic effects [8,9], and developmental toxicity in fish exposed during early life history stages [10,11]. Co-occurrence of toxins and secondary metabolites from cyanobacteria is the environmentally relevant exposure scenario, and research is needed to resolve synergism/antagonism and attribute biological activity to specific substances produced by cyanobacteria. A focus only on cyanotoxins for risk assessment and management of bloom episodes will hinder detection of negative effects on the ecosystem caused by secondary metabolites produced by cyanobacteria.

Therefore, the aims of this study were the following: (i) characterize by LC-HR-QTOF-MS the most common secondary metabolites produced by these two species of cyanobacteria; (ii) quantify microcystin variants by LC-QqQ-MS/MS; and, (iii) determine whether the larvae of the freshwater tropical fish *Astyanax altiparanae* are affected by water-soluble compounds produced by two strains of *Microcystis* (*M. panniformis* MIRS-04—a microcystin producer, and *M. aeruginosa* NPDC-01—a microcystin non-producer.

The two culture collection species of freshwater cyanobacteria had completely distinct metabolite profiles evaluated by comparison of chromatograms, mass spectrometry spectra, as well as by applying the Global Natural Product Social Molecular Networking (GNPS) tools. The results involving the effects of water-soluble extracts from both cyanobacteria on the larval stage of a representative species of neotropical fish *A. altiparanae* provide information relevant to cyanobacterial blooms, the toxicity of other cyanopeptides and their potential for environment impact.

## 2. Results

### 2.1. LC-HR-QTOF-MS Analyses

Figure 1 shows the base peak chromatogram of the most ionizable compounds identified from extracts of strains of *M. panniformis* MIRS-04 (A) and *M. aeruginosa* NPCD-01 (B) in the positive ion mode (ESI^+^). In both extracts, the main abundant compounds eluted between 6 to 10 min; the peaks that appear before and after refer to the conditioning and cleaning of the column (Figure 1). The compounds identified in the extracts were the following: (i) MIRS-04 strain: MC-LR, MC-LY, MC-HilR, micropeptin K139, micropeptin B, micropeptin 959 and micropeptin 973; and (ii) NPCD-01 strain: microginins predominated, microginin FR1, microginin FR2/FR4, microginin SD-755, and a possible variant of cyanopeptolin were found (Table 1).

Cyanopeptides were identified according to their deconvoluted mass spectra, *m*/*z*, molecular formula, exact mass, MS^2^ spectra and also using the GNPS database. Cytoscape was applied for analyzing molecules clustering with a similar fragmentation profile. Cytoscape software integrated main molecules (Table 1) with other peaks and spectra present in the samples, creating groups of compounds with similar characteristics and fragmentation profiles (Figure 2).

MS^2^ spectra of all peaks were further investigated to tentatively characterize the cyanopeptides present in each strain. The MS^2^ spectra of [M + 2H]^2+^ 494.2797 (micropeptin k139), [M + H]^+^
*m*/*z* 959.5249 (micropeptin 959), *m*/*z* 973.5391 (micropeptin 973) and *m*/*z* 987.5548 (micropeptin B) (Appendix A) have similar fragmentation patterns as described by Harada et al. [12] for micropeptins, cyclic structures composed of 3-amino-6-hydroxy-2-piperidone (Ahp), and six amino acid residues: L-arg, L-Ahp, *LN*-MeTyr, L-Asp [13] (Figure 3).

Microcystins are cyclic peptides that present a general structure containing seven amino acid residues including the unusual Adda (2*S*,3*S*,4*E*,6*E*,8*S*,9*S*)-3-amino-9-methoxy-2,6,8-trimethyl-10-phenyldeca-4,6-dienoic acid) [14] (Figure 3). The MS^2^ spectra of the microcystins [M + H]^+^
*m*/*z* 995.5634 (MC-LR), *m*/*z* 1009.5774 (MC-HilR), *m*/*z* 1002.5237 (MC-LY) showed a profile of characteristic fragment ions at *m*/*z* 135 corresponding to the [phenyl-CH_2_CH(OCH_3_)]^+^ from Adda and Glu-Mdha + H (*m*/*z* 213), which are commonly found in microcystins’ MS^2^ spectra [14,15,16] (Appendix A).

The MS^2^ spectra of microginins [M+H]^+^
*m*/*z* 728.4272 (microginin FR1); *m*/*z* 742.4347 (microginin FR2/FR4), *m*/*z* 756.4667 (microginin SD-755) showed an Ahda (3-amino-2-hydroxydecanoic acid) fragment with *m*/*z* 128 and MeAhda (*N*-methyl-3-amino-2-hydroxydecanoic acid) fragment (*m*/*z* 142) (Appendix A), diagnostic ions from the cleavage of Adha either demethylated or the *N*-methylated variant, respectively [17,18,19] (Figure 4). Peak 4 (Figure 1), ion *m*/*z* 960.4188, was not completely identified. The ion assignations for the MS^2^ spectrum for *m*/*z* = 404 [Ahp-Phe-MePhe -H_2_O]^+^, *m*/*z* = 292 [Phe-MePhe -H_2_O]^+^, *m*/*z* = 243 [Ahp-Phe -H_2_O]^+^, *m*/*z* = 215 [Ahp-Phe -H_2_O -CO]^+^, *m*/*z* = 135 [MePhe]^+^, *m*/*z* = 120 (immonium ion typical for Phe) and *m*/*z* = 86 (immonium ion typical for Leu/ILe) (Appendix A), followed the classification suggested by Welker et al. [20], therefore, we suggested this compound to be variant of cyanopeptolin.

### 2.2. MC-LR Quantification by LC-QqQ-MS/MS

The commercially available analytical standards of MC-LR, MC-RR, MC-LA and MC-YR were considered for this study and their detection and quantification were carried out in aqueous extracts of both strains. Only MC-LR was detected and quantified in the *M. panniformis*. The concentrations of MC-LR were 0.15, 0.30, 0.44, 0.59 and 0.74 μg/mL, equivalent to 0.1, 0.2, 0.3, 0.4 and 0.5 mg (dry weight)/mL of MIRS-04 aqueous extract. Microcystins LR, RR, LA and YR were not detected in the strain NPCD-01, including the analyses by LC-HR-QTOF-MS as described above and also by searching the precursor ions corresponding to the losses of *m*/*z* 135 from Adda and *m*/*z* 265 for ADMAdda containing compounds (Appendix A). Therefore, strain NPCD-01 was defined as a MC non-producer.

### 2.3. Acute Toxicity

Exposure conditions (dissolved oxygen 8.0 to 8.6 mg/L) and experiment endpoints (mortality and morphological changes <10% in unexposed controls) were acceptable for interpretation of treatment effects. Larval fish mortality increased with concentration and duration of exposure in fish exposed to both MIRS-04 and NPCD-01 strains. At the 0.4 and 0.5 mg/mL concentrations tested, mortality was greater than 50% within 24 h and above 90% after in 96-h exposure, *p* < 0.05 (Figure 5). LC_50_ values of the MIRS-04 strain (MC producer) were 0.40 (0.36–0.44) mg/mL in 24 h and 0.24 (0.21–0.26) mg/mL in 96 h of exposure time and to NPCD-01 (MC non-producer) strain were 0.42 (0.39–0.44) mg/mL in 24 h and 0.32 (0.28–0.35) mg/mL in 96 h (Figure 5).

### 2.4. Morphological Changes

The morphological abnormalities observed in the larvae of *A. altiparanae* exposed to *Microcystis* ssp. extracts were observed in a significantly higher proportion of fish compared to the control (Table 2). The morphological abnormalities observed were oedema in the pericardium, oedema in the region of the digestive system, curvature of the tail and lordosis, and occurred after exposure to both strains MIRS-04 and NPCD-01 (Table 2). Oedema in the pericardial region (OPC) and oedema in the digestive system (ODS) were observed in the three concentrations of strain MIRS-04 and NPCD-01 (Table 3). The tail curvature was observed at 0.2 mg/mL and 0.3 mg/mL NPCD-01. Oedema in the digestive system was observed at higher frequency than pericardial oedema (*p* < 0.05). 

From the total of observed larvae, >50% presented oedema in the digestive system (ODS) at an exposure concentration of 0.3 mg/mL of the strains MIRS-04 and NPCD-01 *p* < 0.05 (Table 3). The larvae exposed at the concentration of 0.3 mg/mL of MIRS-04 and NPCD-01 presented up to three different types of morphological alterations in the larvae (Figure 6). The results of the morphological alterations also showed that the number of abnormalities in the larvae increased with the concentration of the extracts, *p* < 0.05 (Figure 6 and Figure 7). 

## 3. Discussion

Our results document the presence of distinct peptide profiles produced by *M. panniformis* (MIRS-04) and *M. aeruginosa* (NPCD-01). The linear peptides named as microginins were the most abundant compounds found in NPCD-01, while the MIRS-04 strain presented mainly cyclic variants such as microcystins and micropeptins (Table 1). *M. panniformis* and *M. aeruginosa* are cosmopolitan phytoplanktonic species commonly found in cyanobacterial bloom episodes in South America [24] and are also able to produce microcystins [25,26]. In fact, secondary metabolites from freshwater cyanophytes have been reported in the literature as potent bioactive compounds, including toxic compounds and molecules presenting other mechanisms of action [27].

In order to tentatively identify the most abundant peptides in *Microcystis* species, we used our in-house tandem mass spectral library search to annotate MS/MS spectra from screening cyanometabolites. MS/MS-based annotation of chemical structures acquired with high resolution (i.e., QTOF and Orbitrap) has been increasing in the environmental field [28] and public libraries for natural products can be more easily found nowadays, such as in the GNPS [29]. Cytoscape allowed us to generate molecular network profiles that were exclusive for both strains and illustrated clearly that they are distinct (Figure 2). Although freshwater cyanobacterial metabolites are poorly covered in GNPS, some marine related compounds in environmental marine cyanobacteria were identified using a similar approach [30].

NPCD-01 was already analysed by a direct competitive ELISA based on polyclonal antibodies (Beacon Analytical Systems Inc., Saco, ME, USA) and microcystins were not detected in this strain of *Microcystis aeruginosa* [31]. We also quantified MC-LR in MIRS-04 and water-soluble extracts were prepared based on its concentration by dried weight, varying from 0.15 to 0.74 µg/mL of MC-LR, as well as for NPCD-01 (non-MC-producer) equivalent in mg/mL of dried cells. MS/MS-based annotation revealed in the MIRS-04 extract two other types of microcystins, MC-HilR and MC-LY, both of which were not quantified because there are no commercial analytical standards available. Other peptides such as micropeptins (micropeptin K139, micropeptin B, micropeptin 959 and micropeptin 973) were also identified in extracts of strain MIRS-04. In the analysis of NPCD-01 extract, we identified three microginins (microginin FR1, microginin FR2/FR4 and microginin SS-755) and a possible cyanopeptolin variant (not characterized). We also investigated in the QTOF ESI^+^ extracted ion chromatograms, the presence of ADMAdda + H - CH_3_COOH - NH_3_ (*m*/*z* = 265) and PhCH_2_CH(OMe)^+^ (*m*/*z* = 135) (from Adda), diagnostic ions for microcystins. We did not find any peak corresponding to these ions (Appendix A).

The concentration and distribution of MCs in freshwater lentic systems were quite variable due to the methodologies applied to their detection and quantification [32]. However, high levels of MCs (including intracellular and extracellular) have been reported in different continents during cyanobacterial bloom events and these concentrations range from 0 to 36.5 mg/L [32,33,34].

Cyanobacterial blooms and the presence of several polar metabolites are relatively common and well described worldwide [5,35,36,37]. Usually, cyanotoxins have been carefully investigated in aquatic environments and their toxicity, dynamics, environmental fate, bioavailability, accumulation and magnification are documented [38,39]. Therefore, water-soluble extracts derived from cyanobacteria are a realistic option for studying synergy and toxicological effects of cyanopeptides in aquatic organisms [40].

The synergetic effects among polar secondary metabolites excreted by cyanobacteria during their life stages and senescence, especially the co-existence of toxins and other bioactive peptides in the same environment and similar availability, are only now being investigated in aquatic ecosystems [40]. Most studies reported allelopathy and quorum-sensing proprieties for cyanobacterial metabolite [41,42] interaction with other organisms in water [43,44]. Dynamics of cyanopeptides in the aquatic food web are available in the literature; however, only a few ecotoxicological studies with peptides other than toxins have been published so far [45]. Generally, the toxicity of these compounds seems to be low and in inhibition activity assays, however, some linear and cyclic peptides show extremely low IC_50_ for several key enzymes in the metabolism [27,46,47]. 

Considering these aspects, we evaluated the possible synergy and potential toxicity of different polar metabolites produced by two species of cyanobacteria that were previously classified as microcystin-producer and non-producer on the larva development of *A. altiparanae*, one of the most common fish species in tropical Brazilian water bodies [48,49] *Astyanax altiparanae* is a non-migratory omnivorous fish native from Brazil and the *Astyanax* genus has an abundance of species distributed throughout the Neotropical region [49,50]. This species is an excellent candidate to be used as model for ecotoxicological studies [49,50,51,52,53]. Thus, we chose this species to verify its sensitivity to cyanotoxins, secondary and primary metabolites.

We investigated water-soluble extracts of both *Microcystis* and they showed very similar results for mortality and morphological changes in larvae of *A. altiparanae*. In addition to MCs, especially MC-LR, other substances such as microginins, micropeptins and a possible cyanopeptin might also be toxic in the life cycle of freshwater fish. LC_50_ values for the extracts of MIRS-04 and NPCD-01 were equivalent, and results did not differ statistically between the two species of *Microcystis*. However, the effect of strain MIRS-04 was more pronounced than that of strain NPCD-01, indicating the presence of MCs toxin may increase the toxic effects observed in *A. altiparanae* larvae. Most of the studies relate the concentration of MCs present in crude cyanobacteria extracts to the observed effect. Concentrations range from 0.05 μg/mL to 460 μg/mL and fish species commonly used are *Danio rerio*, *Oncorhynchus mykiss*, *Oryzias latipes*, *Misguruns mizolepis* and *Cyprinus carpio* [54,55,56,57,58]. The relationship between embryo-laval or adult fish exposure to total cyanobacterial metabolites is usually accompanied by lethality, teratogenic effects, developmental retardation, morphological and histopathological changes, as well as behavioral changes in diet or swimming movements [38,59,60,61,62,63]. Consequently, the results can vary significantly, according to the profile of primary and mainly secondary metabolites as well as the fish species (native or exotic) and time of exposure.

Omnivorous fish are most susceptible to the accumulation and transfer of cyanotoxins and other potentially toxic metabolites through food webs [64,65]. However, other substances of cyanobacteria might be equally toxic, as shown in our study. The metabolites produced by the NPCD-01 strain, especially microginins, could cause similar toxicity compared to the MIRS-04 strain, which is able to produce microcystins.

The effect of microginins on aquatic organisms is still poorly studied. Microginins as well as micropeptins and cyanopeptolins are described as bioactive compounds involved in the inhibition of mammalian proteases such as the Angiotensin Converting Enzyme (ACE), resulting in decreased blood pressure [19,23,27]. The biological activity of the *Microcystis* spp biomass containing FR1 microginins was evaluated by Neumann et al. [17] and Forchert et al. [66], showing inhibition of proteases and cytotoxic action in the embryonic development of *Danio rerio*. Considering the morphological changes during exposure, we observed equivalent results for the larvae of *A. altiparanae*. These alterations may have been caused by exposure to microginins. Other effects were reported on Allium plant cells also using the biomass of strain NPCD-01, the same used in the present study. The results showed that the compounds present in NPCD-01 extract, including the microginin variant *m*/*z* 742, caused genotoxic effects, inhibition of root growth and root abnormalities on Allium cells [31]. 

The changes we describe herein include oedema in the abdominal region, mainly in the heart and digestive system. Similar results were also reported in other studies [54,55,58,67,68]. Tissue haemorrhage was noticed in some samples and was probably caused by the presence of MC-LR, LY and HilR variants produced by MIRS 04 strain (Figure 7). Irregular swimming movements triggered by extract exposure may have caused the alterations in the tail and spine curvature of *A. altiparanae* larvae. This behavior was compared to control treatment larvae, which showed normal straight and regular swimming movements (Figure 7). Under stressful conditions, metabolic changes are expected in response to different environmental variations and the development of tissue oedema, size and shape modification are effects previously reported for several aquatic organisms [56]. In the initial stage of fish development, these effects may compromise survival since larvae are more sensitive to toxic compounds in freshwater ecosystems [68,69,70,71,72,73].

## 4. Conclusions

Our study demonstrates that exposure to water-soluble metabolites of cyanobacteria can cause toxicity in developing fish larvae and that effects are not solely caused by exposure to cyanotoxins. Exposure to these secondary metabolites resulted in concentration-related increase in fish mortality and abnormal morphology/development. Characterization of the secondary metabolites produced by cyanobacteria and investigation of their toxicity individually and in combination are essential for understanding the substances responsible for negative effects. Finally, the present study is further confirmation that negative effects of blooms of cyanobacteria are not limited to the toxins they produce, and management of these bloom episodes must consider more than just the concentration of cyanotoxins present in surface waters.

## 5. Materials and Methods

### 5.1. Cultivation of Microcystis Strains

The experiments were conducted using two cyanobacteria strains: *Microcystis panniformis* (MIRS-04) isolated from the Samuel Reservoir in Rondônia, Brazil, and *Microcystis aeruginosa* (NPDC-01) isolated from a Jacarepaguá lagoon in Rio de Janeiro, Brazil. All strains were obtained from the culture collection of the Laboratory of Ecophysiology and Toxicology of Cyanobacteria (LECT), at the Federal University of Rio de Janeiro (UFRJ -Brazil).

The strains MIRS-04 and NPCD-01 were grown in ASM-1 medium, pH adjusted to 7.5, for 20 days [74]. The culture flasks were maintained under a light intensity of 40 μmol photons m^−2^ s^−1^, photoperiod of 12 h, continuous aeration and temperature of 24.0 ± 1.0 °C. After 20 days of growth, the cultures were interrupted and centrifuged (Eppendorf 5804R, BR) at 10,000 RPM for 10 min at 4 °C. The resulting biomass was stored at −20 °C and then lyophilized. Lyophilized cells were used in the analyses of the secondary metabolites by LC-HR-QTOF-MS, quantification of MC-LR by LC-QqQ-MS/MS and for acute toxicity tests with *A. altiparanae* larvae.

### 5.2. Extraction Procedure

The extraction of microcystins and cyanopeptides from the strains of *Microcystis* ssp. followed the freeze-thaw protocol with some adaptations of methods described in the literature that are usually used to extract water-soluble cyanotoxins and other substances from cyanobacteria [38]. Stock solutions were prepared from 100 mg of MIRS-04 and NPCD-01 dry cells diluted in 20 mL of Milli-Q water with the addition of 12 mM NaCl, 1 mM KCl, 1.5 mM CaCl_2_, and 1.5 mM MgCl_2_ (reconstituted water used for the development of *A. altiparanae*). The extracts were frozen at −20 °C and thawed at 23 °C, freeze-thaw cycles were performed for 6 consecutive times with 3 h intervals. The broken cells were stored at −20 °C until analysis for secondary metabolite identification, quantification of MCs and the acute toxicity tests with *A. altiparanae*.

### 5.3. LC-HR-QTOF-MS—Analyses

The samples used in this analysis were prepared from stock solutions of the MIRS-04 and NPCD-01 strains and dilutions of the samples were performed with reconstituted water. Solutions with 5 mg/mL were used for both strains. The extracts were centrifuged at 11.000 RPM, 4 °C for 10 min and the supernatants collected and filtered on 0.22 μm PVDF membranes (Millipore, Bedford, MA, USA) [36]. The *M. panniformis* (MIRS-04) and *M. aeruginosa* extracts (NPCD-01) were analyzed using an ultra-high performance liquid chromatograph (Shimadzu, Kyoto, Japan) coupled to a hybrid quadrupole time-of-flight high resolution mass spectrometer (LC-HR-QTOF-MS) (Compact II, Bruker Daltonics Corporation, Germany) equipped with an electrospray ionization (ESI) source in positive mode. Chromatographic separation was performed with a Trio Column C18 packed with 2.0 μm particle size, 2 × 100 mm column (YMC Technology, Allentown, PA, USA) at a flow rate of 0.4 mL/min.

The gradient mixture of solvents A (H_2_O with 0.1% formic acid; *v*/*v*) and B (acetonitrile: H_2_O [90:10] with 0.1% formic acid; *v*/*v*) (Sigma-Aldrich, Darmstadt, Germany), was the following: from 5% to 95% B in 10 min, maintained at 95% B 10–12 min, 5% B 12–12.5 min, and maintained at 5% B 12.5–15 min at 40 °C. The capillary voltage was operated in the positive and negative ionization modes, set at 4500 and 3000 V, respectively; with an end plate offset potential of −500 V. The dry gas parameters were set to 9 L min/^−1^ at 200 °C with a nebulization gas pressure of 5 bar.

Data were collected from *m*/*z* 60 to 2000 with an acquisition rate of 5 Hz, and the variable number of ions was selected by auto MS/MS scan fragmentation with a cycle time of 2.5 seconds. The MS/MS data were analyzed using Bruker Compass Data Analysis 4.3 software, and it was possible to calculate the experimental mass of each compound [M + H]^+^ and [M + 2H]^2+^, the molecular formula, the error in ppm <5 and millisigma values (mSigma) < 20.

### 5.4. Molecular Networking

The Molecular Network (MN) approach required the conversion of mass spectrometry raw data into mzXML file format followed by upload to the Global Natural Products Social Molecular Networking site (GNPS—https://gnps.ucsd.edu) [29]. The MN was generated so that edges had a cosine score above 0.65 and six matched peaks. Edges between two nodes were kept in the MN if each of the nodes appeared in each other’s respective top 10 most similar nodes and then clustering with MS-Cluster with parent mass and MS/MS fragment ion tolerances of 0.02 Da. Consensus spectra that contained less than two spectra were discarded. The spectra in the MN were then searched against GNPS spectral libraries. The library spectra were filtered in the same manner as the input data. All matches between network spectra and library spectra were required to have a score above 0.65 and at least six matched peaks. The MN data was downloaded and visualized in Cytoscape [75].

### 5.5. Liquid Chromatography–Tandem Mass Spectrometry LC-QqQ-MS/MS Analyses

The quantification of MCs in aqueous extracts of the MIRS-04 and NPCD-01 strains was performed by LC-QqQ-MS/MS using a method previously validated by our group [76]. The extracts used in this analysis were also prepared from stock solutions of the MIRS-04 and NPCD-01 strains at a concentration of 5 mg/mL. The major microcystin variants MC-LR, MC-RR, MC-LA and MC-YR (Abraxis, Warminster, PA, USA) were used to determine the concentration of MCs in the extracts. Analytical standards were prepared in MilliQ water at concentrations of 0.1 µg/ml to 5 µg/mL. Identification and quantification of MC in the MIRS-04 strain extract were determined by tandem MS/MS analysis performed in multi-Reaction Monitoring Mode (MRM), monitoring the ionic products of *m*/*z* 135 and 213 common to the variants of microcystins that are most commonly found in the environment.

A calibration curve was constructed with MC solutions ranging from 0.1, 0.5, 2.0, 3.0, 5.0, µg/mL of mixed MCs. MC-LR was identified by comparing the sample retention time of the extracts with the retention time of the analytical standard. The concentration of MC-LR in the MIRS-04 strain was determined from the linear relationship between the areas of the precursor ion [M + H]^+^ of *m*/*z* 995.5 corresponding to the protonated molecule, which also produces *m*/*z* [M + H]^+^ 135 and 213.

The samples were analyzed and quantified in an Agilent 1260 Infinity chromatographic system coupled to a triple quadrupole mass spectrometer (6460 Triple Quadruplo LC-MS, Agilent Technologies, Santa Clara, CA, USA), with ionization by electrospray (ESI), in positive mode at 3500V. Nitrogen was used as the gas nebulizer (45 psi) and drying gas (5 mL/min at 300 ° C). Separation of the compounds was performed on a Synergi Fusion-RP, 4 μm, 80 Å, 150 × 2.0 mm column (Phenomenex, Torrance, CA, USA) under linear gradient. The gradient mixture of solvents **A** (H_2_O with 0.1% formic acid; *v*/*v*) and **B** (acetonitrile:H_2_O [90:10], with 0.1% formic acid; *v*/*v*) (Sigma-Aldrich, Darmstadt, Germany), both containing 2mM of ammonium formate at a flow rate of 0.25 mL min, was as follos: from 35% to **B** in 10 min, maintained at 50% **B** 10–12 min, 100% **B** 12–13 min, and maintained at 35% **B** 14–20 min at 40 °C. The volume of injection used for the analyses with analytical standards of microcystins and extracts of strain MIRS-04 were 5 μL, at the different concentrations.

### 5.6. Samples Preparation for the Toxicity Tests

The concentrations used in the assays were prepared from the dry weight of strain MIRS-04 and NPCD-01, diluted in reconstituted aqueous medium (12 mM NaCl, 1 mM KCl, 1.5 mM CaCl_2_, 1.5 mM MgCl_2_) as described in item 5.2 Extraction procedure. The extract concentrations of the MIRS-04 and NPCD-01 strains were established from initial tests on logarithmic scales. After this procedure, the find concentrations 0.1, 0.2, 0.3, 0.4 and 0.5 mg/mL were used for both assays with the Microcystis strains.

### 5.7. Astyanax Altiparanae Larvae Acquisition

The experiments were conducted at the National Center for Research and Conservation of Continental Fish (CEPTA) of the Chico Mendes Institute of Conservation and Biodiversity (ICMBio) Pirassununga, São Paulo State, Brazil, between October and February (2015/2016). This study was approved on 12th of June of 2015 by the ethics committee of CEPTA for the use of specimens of *A. altiparanae* according to the Normative Resolutions of the National Council for the Control of Animal Experimentation (CONCEA) protocol number 007/2015. Adult specimens of *A. altiparanae* were collected in the Mogi Guaçu River (21°58′S, 47°26′W), in Pirassununga, São Paulo, Brazil.

The broodstock were kept in tanks (11.25 m^3^) with constant water flow and fed twice a day with commercial pellets. After a few weeks, the fish were selected for reproduction based on external morphology (females presenting increased volume of the peritoneal cavity and males present tiny hooks in the anal fin). Different pairs of fish were used to ensure the genetic variability of the group of *A. altiparanae* larvae evaluated in the acute toxicity tests. After approximately 10 h, the fish were anesthetized in 100 mg/L of menthol (Exodus Sciencia, São Paulo, Brazil). In the females, injections were applied intraperitoneally under the caudal fin at the dose of 3 mg/kg of hormone (OVOPEL^®^ [(D-Ala^6^,Pro^9^-Net)-mGnRH+metaclopramide]) [77,78,79].

After the hormonal inductions, brood stock were returned to the aquarium for breeding. After 6 h of breeding, all eggs were collected from the aquarium and divided into glass Petri dishes (100 mm diameter) coated with bovine serum albumin 0.1% (Sigma-Aldrich, Darmstadt, Germany) with reconstituted water medium. During embryonic development, water was renewed at every two hours and the dead embryos were removed immediately to avoid ammonia contamination. Incubation took place in Petri dishes, which were placed in a biochemical oxygen demand (BOD) incubator with temperature set at 26 °C until toxicity trials. 

### 5.8. Acute Effect in Larvae of Astyanax Altiparanae

The experiments were performed with larvae 2 h post hatching, with normal morphological characteristics, such as uniform body, motility, constant heart beat frequency and yolk sac under normal conditions. The assays consisted of five different concentration treatments (dried weight) with extracts from strain MIRS-04 and NPCD-01 and control (only reconstituted water). The larvae were exposed in test tubes with a capacity of 10 mL, in the proportion of one larva/mL. Five replicates were performed, each replicate with 10 larvae per treatment (50 larvae per concentration). All treatments were maintained under the same conditions in a BOD incubator, with temperature set at 26 °C and 12 h of light. An aliquot of the samples used in the two assays was separated to measure the initial and final dissolved oxygen concentration with the aid of a model 5100 YSI oxygen meter (YSI, Yellow Springs, OH, USA).

Observations were made every 24 h from 24 to 96 h determine mortality and to evaluate morphological alterations. Morphology and early development were observed with a Nikon SMZ 1500 stereomicroscope (Nikon, Tokyo, Japan), and digital images obtained with a Ds-F1 CCD camera (Nikon, Tokyo, Japan) with the Nis-Ar Elements software (Nikon, Tokyo, Japan). At each observation, the dead larvae were counted and removed from the dishes in order to maintain the water quality. During these observations, the apparent abnormalities in the surviving larvae of all treatments were also registered for further morphological analysis, such as pericardial oedema, oedema in the digestive system region, curvature of the tail and spine, and changes in body size.

### 5.9. Statistical Analyses

Nonlinear regression analysis was used to estimate the lethal concentration of 50% the of total population (LC_50_) of the extracts of *M. aeruginosa* (NPCD-01) and *M. panniformis* (MIRS-04) on the larvae of *A. altiparanae* using GraphPad Prism Version 7.0 software. The other statistical analyses were performed with the statistical software R version 3.1.2 [80]. To evaluate the difference in mortality of *A. altiparanae* larvae between the different concentrations of MIRS-04 and NPCD-01 strains, analysis of variance analysis (ANOVA) was performed, followed post hoc by Tukey’s test. The morphological alteration data were analysed by the chi-square test (χ^2^) to verify which of the morphological alteration variables were more representative among the treatments with the extracts of *Microcystis* ssp. Next, generalized linear regression with a Poisson distribution was performed to test the hypothesis that higher concentrations of the extracts influence the number of morphological changes per larva. The graph of these data was plotted with GraphPad software.

## Figures and Tables

**Figure 1 toxins-11-00220-f001:**
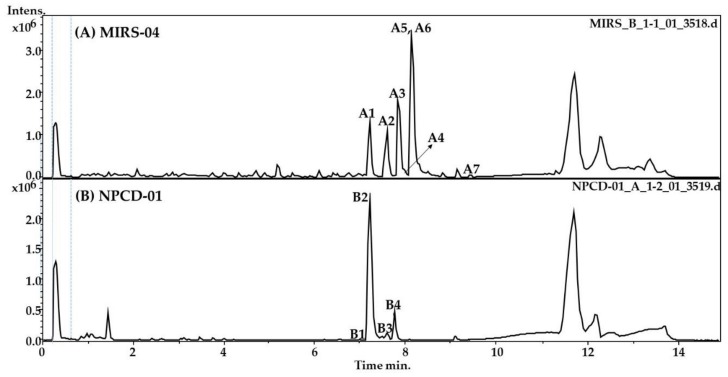
Representative base peak chromatograms detected in positive mode electrospray ionization (ESI), acquired in a LC-HR-QTOF-MS (60-2000 *m*/*z*). *Microcystis panniformis* (MIRS-04): (**A1**) micropeptin K139, (**A2**) micropeptin 959, (**A3**) microcystin-LR, (**A4**) microcystin HiLR, (**A5**) micropeptin 973, (**A6**) micropeptin and (**A7**) microcistin-LY.and *Microcystis aeruginisa* (NPCD-01): (**B1**) microginin FR1, (**B2**) microginin FR2/FR4, and (**B3**) microginin SD-755 and (**B4**) unknown.

**Figure 2 toxins-11-00220-f002:**
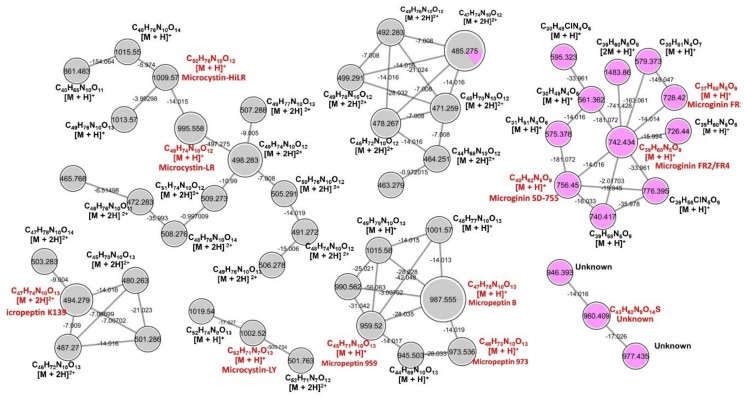
Molecular network generated from the MS/MS spectra of *Microcystis panniformis* (MIRS-04) and *Microcystis aeruginisa* (NPCD-01) strains using GNPS and Cytoscape tools. Clusters of the analytes detected in *M. panniformis* and *M. aeruginosa* are indicated in gray and pink, respectively. Each cluster allows a visualization of related molecules where one node represents one consensus MS/MS spectrum and is labeled with the precursor mass. Circle diameter of each node is proportional to intensity of the ions in the LC-MS analysis. An edge represents relatedness (cosine similarity score cutoff of 0.65) where numbers indicate the mass difference. For each node, calculated molecular formula and ions charges states are informed. Identified compounds are labeled in red while nonidentified nodes indicate new potential analogues. Node 485.275 [M + 2H]^2+^ was detected in both species and ion intensity in each specie is represented as a pizza plot.

**Figure 3 toxins-11-00220-f003:**
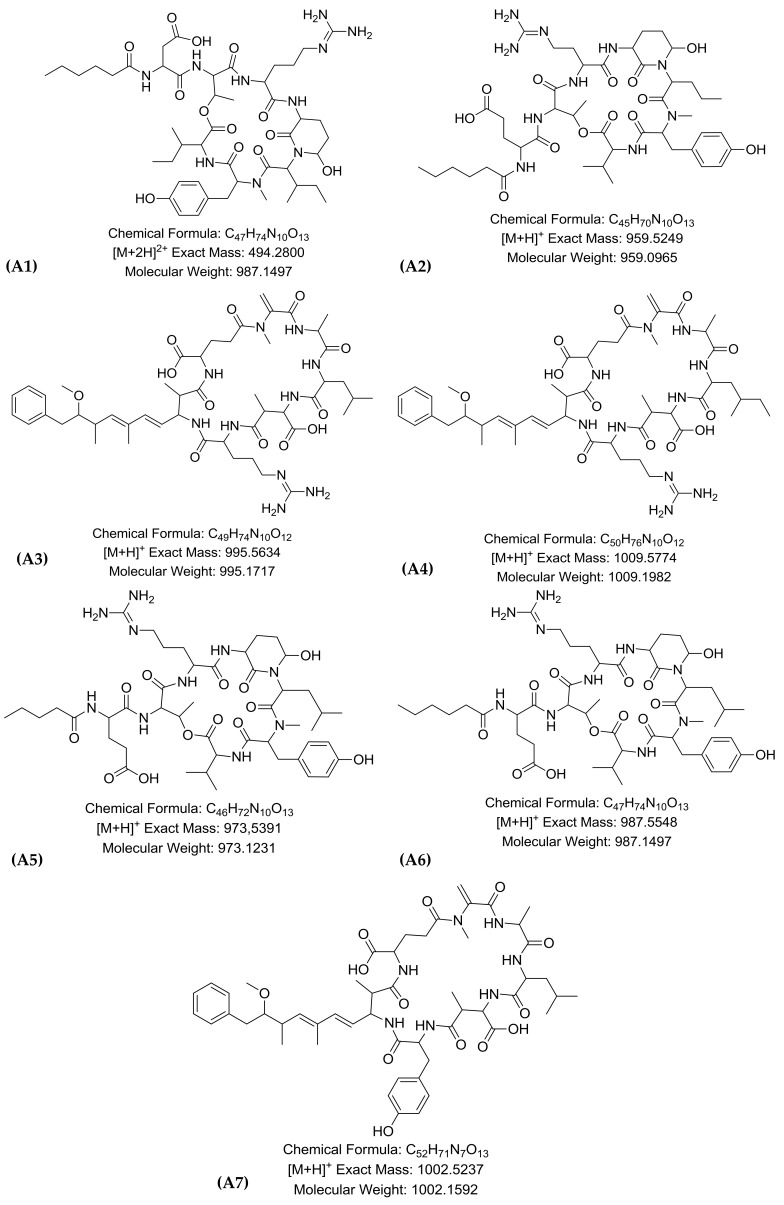
Cyclic peptide structures produced by *M. panniformis* strain MIRS-04: (**A1**) micropeptin K139, (**A2**) micropeptin 959, (**A3**) microcystin-LR, (**A4**) microcystin HiLR, (**A5**) micropeptin 973, (**A6**) micropeptin B and (**A7**) microcistin-LY.

**Figure 4 toxins-11-00220-f004:**
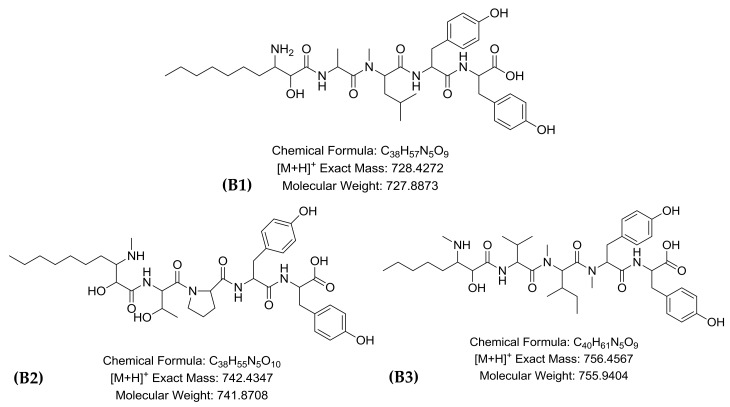
Microginins produced by *M. aeruginosa* strain NPCD-01: (**B1**) microginin FR1, (**B2**) microginin FR2/FR4 and (**B3**) microginin SD-755.

**Figure 5 toxins-11-00220-f005:**
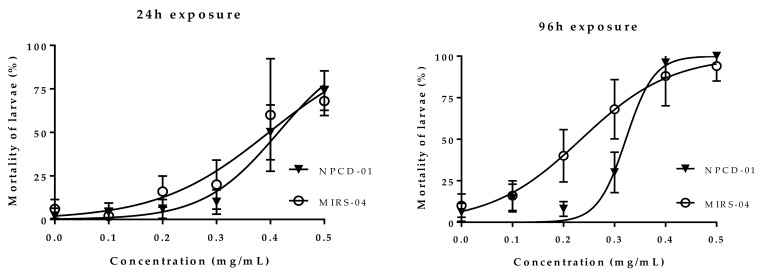
Mortality of *Astyanax altiparanae* larvae (%) vs. concentration (mg/mL) of extracts of strain MIRS-04 and NPCD-01. Data are expressed as dose-response curves as ± SD of five replicates of the assays (MIRS-04 and NPCD-01) and the LC_50_ values are represented for exposure times of 24 and 96 h. (Statistically significant values are at the *p* < 0.05 level).

**Figure 6 toxins-11-00220-f006:**
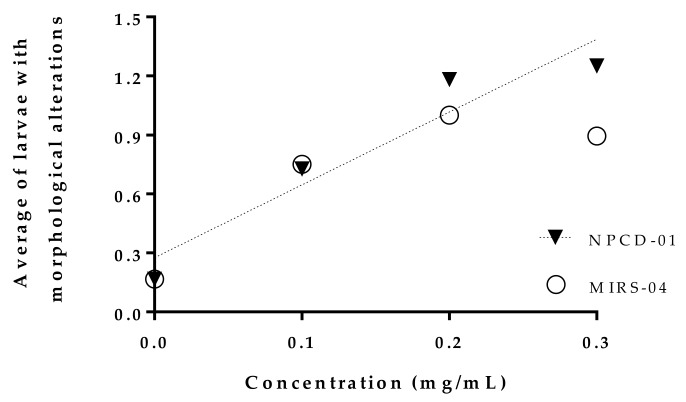
Average of the number of morphological alterations in larvae of *Astyanax altiparanae* exposed to MIRS-04 (*Microcystis panniformis*) and NPDC-01 (*Microcystis aeruginosa*).

**Figure 7 toxins-11-00220-f007:**
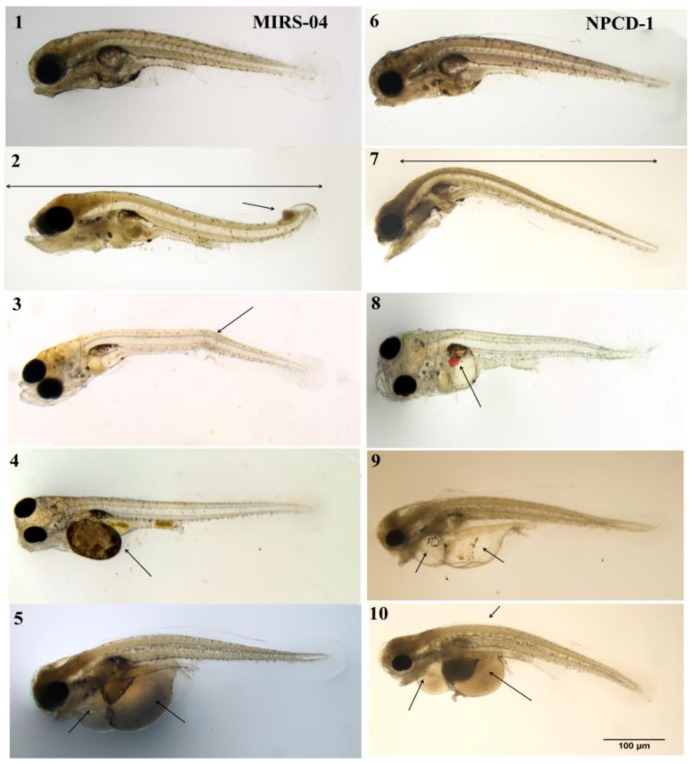
Morphological changes in *Astyanax altiparanae* larvae caused by exposure to extracts of the MIRS-04 and NPCD-01 strains at concentrations of 0.1, 0.2 and 0.3 mg/mL (dried weight). MIRS-04: 1—normal larvae (control), 2—alteration in the tail, column and body length, 3—spinal curvature, 4—digestive system oedema, and 5—oedema in the pericardium and digestive system oedema. NPCD-01: 6—normal (control) larvae, 7—curvature of the spine (cervical lordosis), 8—oedema with haemorrhage, 9—oedema in the pericardium and digestive system oedema, 10—oedema in the pericardium, digestive system oedema and curvature in the cervical spine.

**Table 1 toxins-11-00220-t001:** Cyanobacteria peptides from the strains MIRS-04 and NPCD-01 obtained from the LC-HR-QTOF-MS analyses.

Peak	RT *	Theoretical *m*/*z*	Experimental *m*/*z*	Ion Formula	Error [ppm]	mSigma	Compound Name	Reference
A1	7.2	494.2800	494.2797	C_47_H_76_N_10_O_13_	−0.6	2.3	Micropeptin K139 *	[12]
A2	7.6	959.5249	959.5197	C_45_H_71_N_10_O_13_	−2.0	10.0	Micropeptin 959	[21]
A3	7.9	995.5634	995.5560	C_49_H_75_N_10_O_12_	−0.3	16.5	Microcystin–LR	[16]
A4	8	1009.5774	1009.5717	C_50_H_77_N_10_O_12_	−0.7	20	Microcystin HilR	[14]
A5	8.0	973.5391	973.5353	C_46_H_73_N_10_O_13_	4.1	8.6	Micropeptin 973	[13]
A6	8.2	987.5548	987.5510	C_47_H_75_N_10_O_13_	−0.8	11.4	Micropeptin B	[12]
A7	9.5	1002.5237	1002.5183	C_52_H_72_N_7_O_13_	2.1	5.8	Microcystin-LY	[22]
B1	7.0	728.4272	728.4229	C_38_H_58_N_5_O_9_	5.2	14	Microginin FR1	[17]
B2	7.2	742.4347	742.4379	C_39_H_60_N_5_O_9_	0.7	1.9	Microginin FR2/FR4	[20]
B3	7.6	756.4567	756.4542	C_40_H_62_N_5_O_9_	−3.2	5.3	Microginin SD-755	[23]
B4	7.8	960.4188	960.4131	C_43_H_62_N_9_O_14_S	3.5	2.0	Unknown	[20]

* Detected as [M + 2H]^2+^. All other compounds were detected as [M + H]^+^. **RT**: Retention Time.

**Table 2 toxins-11-00220-t002:** Frequency of occurrence of morphological changes observed in the larvae of *Astyanax altiparanae* in 96 h of exposure to cyanobacteria extracts of MIRS-04 *Microcystis panniformis* and NPCD-01 *Microcystis aeruginosa.*

Treatments	N Total of Larvae	N Total of Larvae M.A (+) *	Types of Morphological Alterations * Observed							
			ODS %	ODS%	OPC%	OPC%	CP%	CP%	CT%	CT%
Control	24	3	3	12.5	1	4.2	0	0	0	0
0.1 mg/mL-mirs-04	28	16	12	42.9	4	14.3	6	21.4	0	0
0.2 mg/mL-mirs-04	19	14	7	36.8	4	21.1	5	26.3	1	5.3
0.3 mg/mL-mirs-04	11	10	9	81.8	2	18.2	2	18.2	0	0
0.1 mg/mL-npcd-01	14	11	11	78.6	0	0	2	14.3	0	0
0.2 mg/mL-npcd-01	22	13	10	45.5	3	13.6	2	9.1	0	0
0.3 mg/mL-npcd-01	24	22	22	91.7	3	12.5	4	16.7	1	4.2
**Total**	142	89	74	52.1	17	12	21	14.8	2	1.4

* M.A (+): Morphological alteration, ODS: oedema in the region of the digestive system, OPC: oedema of the pericardial region, CP: Curvature of the Spine, CT: Curvature of the Tail.

**Table 3 toxins-11-00220-t003:** Chi-square test of the morphological alterations oedema in the digestive system according to presence and absence in the larvae of *Astyanax altiparanae*.

Treatments	ODS (+) *	ODS (−) *	N Total of Larvae	Frequency of Occurrence (%)
Control	3	21	24	12.5 ^a^
0.1 mg/mL-MIRS-04	12	16	28	42.9 ^b, f, g^
0.2 mg/mL-MIRS-04	7	12	19	36.8 ^c, d, e, g^
0.3 mg/mL-MIRS-04	9	2	11	81.8 ^c, d^
0.1 mg/mL-NPCD-01	11	3	14	78.6 ^c, d, e^
0.2 mg/mL-NPCD-01	10	12	22	45.5 ^f, g^
0.3 mg/mL-NPCD-01	22	2	24	91.7 ^b, c, f, g^

* ODS (+): Presence of oedema in the region of the digestive system, ODS (−) absence of oedema in the region of the digestive system. The values with equal letters did not present significant differences *p* < 0.05.

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
