# Peer review of "Toxicity of Cyanopeptides from Two Microcystis Strains on Larval Development of Astyanax altiparanae"

_toxins, 2019, doi:10.3390/toxins11040220_

Round 1
Reviewer 1 Report
In the revision R1 of their manuscript, the authors have now concidered the previous main concerns about the characterization of a solvant extract by mass spectrometry, and the use of water extract for the larvae exposure. The MS seems now almost Ok, and I have only few minor concerns that aim at being considered before formal acceptance for publication.
Minor concerns:
- the highest test concentration is about 740 microg/L Microcystins, please discuss the representativity of such concentration according to envrinmentally realistic conditions.
- Figure 2 : the resolution of figure should be improved. I do not understand what the size of the circle represents. Moreovre, the red circle are nor all rounded, and I recommand to simply and more intuitively represent these metabolites network with a circle diamater that would be related to the number of spectra or the intesnity of the ions.
- Figure 1: remore "al" at the end of the legend.
- L234 "Cyanopeptolin"
- Peak 4: please provide the list of diagnostic fragmentation ions that have been observed and corresponding references, or remove this potential identification
- Please, specify that Micropeptins are Aeruginosins, and discuss the potential bioactivity known for this metabolite family.
Author Response
Authors’ replies to Reviewer 1COMMENTS AND SUGGESTIONS FOR AUTHORS
1. In the revision R1 of their manuscript, the authors have now considered the previous main concerns about the characterization of a solvent extract by mass spectrometry, and the use of water extract for the larvae exposure. The MS seems now almost Ok, and I have only few minor concerns that aim at being considered before formal acceptance for publication.
Authors’ Answer: We would like to thank reviewer 1 for his/her comments and revision of our manuscript. Certainly, considering these modifications, our manuscript will be of more interesting for readers.
Minor concerns:
(i) the highest test concentration is about 740 microg/L Microcystins, please discuss the representability of such concentration according to environmentally realistic conditions.
Authors’ Answer: This question is relevant and very important. We modified the text to justify this level of microcystin (between lines 222 to 226 and 256 to 258). The choice of dose levels was based on information available in literature about high levels of microcystins in aquatic environment during bloom events. In our particular case, we found similar concentration in a previous water monitoring (Bortoli et al 2014, cited in our manuscript) and, more recently, studies published by Menezes et al. 2017 and Turner et al. 2018, that found the total microcystin content of 506 μg/L and 611 μg/L [1,2], respectively.
Also, dissolved (extracellular) concentrations of microcystins may vary from 0.1 to 1800 µg/L in natural water bodies including fishponds [3].
(ii) Figure 2: the resolution of figure should be improved. I do not understand what the size of the circle represents. Moreover, the red circle are nor all rounded, and I recommend to simply and more intuitively represent these metabolites network with a circle diameter that would be related to the number of spectra or the intensity of the ions.
Authors’ Answer: We apologize for that. Resolution was enhanced and added in the manuscript. The notations were modified accordingly in order to make this figure and its interpretation clearer. The legend was improved to accommodate the aforementioned suggestion.
(iii) Figure 1: remove "al" at the end of the legend.
Authors’ Answer: It was deleted. Thanks.
(iv) L234 "Cyanopeptolin"
Authors’ Answer: This misspelled word has been corrected in the text.
(v) Peak 4: Please provide the list of diagnostic fragmentation ions that have been observed and corresponding references, or remove this potential identification
Authors’ Answer: Thanks for this suggestion. We added in the text possible fragmentation ions as well as the references concerning the potential identification of cyanopeptolins by mass spectrometry based on Welker et al (2006) [4]. The ion assignations for m/z = 404 [Ahp-Phe-MePhe -H2O]+, m/z = 292 [Phe-MePhe -H2O]+, m/z = 243 [Ahp-Phe -H2O]+, m/z = 215 [Ahp-Phe -H2O -CO]+, m/z = 135 [MePhe]+, m/z = 120 (immonium ion typical for Phe) and m/z = 86 (immonium ion typical for Leu/ILe) were included in the text. However, we were not able to tentatively identify this peptide.
(vi) Please, specify that Micropeptins are Aeruginosins, and discuss the potential bioactivity known for this metabolite family.
Authors’ Answer: Thank you for this observation, we defined in the text. However, they are not the same class of compounds. According to Welker et al. (2006) [4] aeruginosins are a family of linear peptides with more than 20 members and are able to inhibit the activity of serine proteases such as trypsin, chymotrypsin and thrombin [5]. This class of linear peptides present the exotic 2-carboxy-6-hydroxyoctahydroindole (Choi) moiety as well as the C-terminal arginine derivatives such as argininal, argininol, agmatine and 1-amidino-2-ethoxy-3-aminopiperidine. The N-terminus typically consists of either hydroxyphenyl lactic acid in Microcystis or phenyl lactic acid in Planktothrix [6]. On the other hand, micropeptins have a characteristic unit, 3-amino-6-hydroxy-2-piperidone (Ahp), and a cyclic structure with six amino acid residues. Therefore, structurally, this group has more proximity to the cyanopeptides. The only similarity they have is the ability to inhibit proteases [7]. We clarified in the text in order to make these differences clearer.
REFERENCES
1. Turner, A. D.; Dhanji-Rapkova, M.; O’Neill, A.; Coates, L.; Lewis, A.; Lewis, K. Analysis of Microcystins in Cyanobacterial Blooms from Freshwater Bodies in England. Toxins (Basel). 2018, 10, doi:10.3390/toxins10010039.
2. Menezes, C.; Churro, C.; Dias, E. Risk Levels of Toxic Cyanobacteria in Portuguese Recreational Freshwaters. Toxins (Basel). 2017, 9, doi:10.3390/toxins9100327.
3. Kohoutek, J.; Adamovský, O.; Oravec, M.; Šimek, Z.; Palíková, M.; Kopp, R.; Bláha, L. LC-MS analyses of microcystins in fish tissues overestimate toxin levels—critical comparison with LC-MS/MS. Anal. Bioanal. Chem. 2010, 398, 1231–1237, doi:10.1007/s00216-010-3860-z.
4. Welker, M.; Maršálek, B.; Šejnohová, L.; von Döhren, H. Detection and identification of oligopeptides in Microcystis (cyanobacteria) colonies: Toward an understanding of metabolic diversity. Peptides 2006, 27, 2090–2103, doi:https://doi.org/10.1016/j.peptides.2006.03.014.
5. Ersmark, K.; Del Valle, J. R.; Hanessian, S. Chemistry and Biology of the Aeruginosin Family of Serine Protease Inhibitors. Angew. Chemie Int. Ed. 2008, 47, 1202–1223, doi:10.1002/anie.200605219.
6. Fewer, D. P.; Jokela, J.; Paukku, E.; Österholm, J.; Wahlsten, M.; Permi, P.; Aitio, O.; Rouhiainen, L.; Gomez-Saez, G. V; Sivonen, K. New Structural Variants of Aeruginosin Produced by the Toxic Bloom Forming Cyanobacterium Nodularia spumigena. PLoS One 2013, 8, 1–10, doi:10.1371/journal.pone.0073618.
7. Kisugi, T.; Okino, T. Micropeptins from the Freshwater Cyanobacterium Microcystis aeruginosa (NIES-100). J. Nat. Prod. 2009, 72, 777–781, doi:10.1021/np800631t.
8. Laughinghouse, H. D.; Prá, D.; Silva-Stenico, M. E.; Rieger, A.; Frescura, V. D. S.; Fiore, M. F.; Tedesco, S. B. Biomonitoring genotoxicity and cytotoxicity of Microcystis aeruginosa (Chroococcales, Cyanobacteria) using the Allium cepa test. Sci. Total Environ. 2012, 432, 180–188, doi:10.1016/j.scitotenv.2012.05.093.
Reviewer 2 Report
I'm a natural product chemist that specialises in cyanobacterial secondary metabolites and dabble in the ecotoxicology of these compounds. I don't, however, have a great deal of experience in the latter.
This study assesses the effect of two Microcystis extracts on tropical fish larvae (Astyanax altiparanae). One extract contained the cyanotoxin microcystin whilst the other did not. Mortality of Astyanax altiparanae larvae was assessed, as well as growth abnormalities. The authors found that both extracts had effects.
Overall, I found the study to be simple but thorough. Because of its simplicity I feel that the work would be better suited for presentation in a more succinct form such as a Short Communication. To do this, the author's might cut back the amount of space devoted to characterising secondary metabolites in the extracts and focus more on the results of the ecotoxicology assay.
Personally, I found that the authors place too much emphasis on the secondary metabolites identified in the extracts being the main effectors of toxicity for the fish larvae. But since there were different profiles of identified metabolites in each extract and both extracts produced negative effects for the fish larvae it suggests to me that something else is causative agent. Whilst revising the manuscript, the authors should pay attention to this aspect and the manuscript would likely benefit from some discussion
Since the authors went to the effort of quantifying one of the microcystins present in the extract, I think that there should be more discussion on the differences/similarities between previous studies assessing the effects of microcystins on fish larvae and the results from the microcystin-containing extract assessed in the present study; e.g., were the growth abnormalities observed similar?
Other specific comments:
Throughout - It would make more sense to me if the MIRS-04 strain produced MC-LY rather than MC-YL. The reason I believe this is because the strain is able to incorporate Leu and Hil (or maybe Hle) into Position 2, it would make more sense for the third MC congener to contain a Position 2 Leu rather than Tyr. Tandem MS analysis would identify this.
L25 - Because crude extracts are used, you're assessing the effects of more than just secondary metabolites.
L124 - "Microcystins LR, RR, LA and YR were not detected in the strain NPCD-01, including the experiment by LC-HR‐QTOF‐MS as described above. Therefore, strain NPCD-01 was defined as a MC non-producer" This line is unjustified since there are many other microcystin congeners besides the four listed. Using an Adda ELISA or a less specific LC-MS technique such precurser ion scanning (https://www.mdpi.com/2072-6651/10/4/147/htm) would be a more believable way to identify the absence of MCs in the extract.
Author Response
Authors’ replies to Reviewer 2COMMENTS AND SUGGESTIONS FOR AUTHORS
1. I'm a natural product chemist that specialises in cyanobacterial secondary metabolites and dabble in the ecotoxicology of these compounds. I don't, however, have a great deal of experience in the latter.
Authors’ Answer: It is our pleasure to have our manuscript revised by an expert in the field of cyanobacterial metabolites. Comments from Referee #2 were very important and useful for us in order to improve and correct mistakes in our manuscript. We thank very much Referee #2 for his/her deep revision and comments/suggestions.
2. This study assesses the effect of two Microcystis extracts on tropical fish larvae (Astyanax altiparanae). One extract contained the cyanotoxin microcystin whilst the other did not. Mortality of Astyanax altiparanae larvae was assessed, as well as growth abnormalities. The authors found that both extracts had effects.
Authors’ Answer: This is an excellent summary of the present work.
3. Overall, I found the study to be simple but thorough. Because of its simplicity I feel that the work would be better suited for presentation in a more succinct form such as a Short Communication. To do this, the author's might cut back the amount of space devoted to characterising secondary metabolites in the extracts and focus more on the results of the ecotoxicology assay.
Authors’ Answer: We agree with Referee #2 that some points of the manuscript may be relatively simple. On the other hand, we emphasize that our findings are innovative for not only the species used, a tropical fish, but also the toxicological aspects regarding other metabolites produced by cyanobacteria in addition to microcystins. Furthermore, our work used accurate analyses and adequate experimental design that are in line with the requisites of “Toxins” journal. Based on our comments above, we would like to maintain the present structure for publication, however, if the Editor recommends a “Short communication”, we may adequate for the new structure. We hope that Referee#2 may be comprehensive in this regard.
4. Personally, I found that the authors place too much emphasis on the secondary metabolites identified in the extracts being the main effectors of toxicity for the fish larvae. But since there were different profiles of identified metabolites in each extract and both extracts produced negative effects for the fish larvae it suggests to me that something else is causative agent. Whilst revising the manuscript, the authors should pay attention to this aspect and the manuscript would likely benefit from some discussion.
Authors’ Answer: We completely concur. This is one of the main innovative aspects of this work, which emphasized not only the microcystin per se, but also other metabolites that also present deleterious effects on fish, as observed in our data. We rearranged the text in order to emphasize such observation.
5. Since the authors went to the effort of quantifying one of the microcystins present in the extract, I think that there should be more discussion on the differences/similarities between previous studies assessing the effects of microcystins on fish larvae and the results from the microcystin-containing extract assessed in the present study; e.g., were the growth abnormalities observed similar?
Authors’ Answer: Thank you for the suggestion, we added extra information about levels of microcystins in natural environments and during bloom episodes as well as effects of other cyanobacterial metabolites in freshwater fish population. We modified the text adding levels of microcystins (between lines 222 to 226 and 256 to 258) and discussion.
Other specific comments:
(i) Throughout - It would make more sense to me if the MIRS-04 strain produced MC-LY rather than MC-YL. The reason I believe this is because the strain is able to incorporate Leu and Hil (or maybe Hle) into Position 2, it would make more sense for the third MC congener to contain a Position 2 Leu rather than Tyr. Tandem MS analysis would identify this.
Authors’ Answer: Absolutely right. We double checked the fragmentation pattern and MC-LY is the correct structure. The ions assignation m/z = 293 [Tyr-MeAsp + H]+, m/z = 397 [Mdha-Ala-Leu-MeAsp + H]+ and m/z = 423 [Tyr-MeAsp-Leu -NH2 + 2H]+ tentatively indicate that Leu is in the position 2 and Tyr is in the position 4. The correct structure and sequence of this MC variant was changed in the text (MC-LY was added instead).
(ii) L25 - Because crude extracts are used, you're assessing the effects of more than just secondary metabolites.
Authors’ Answer: We fully agree. There are more than secondary metabolites. Water-soluble pigments, proteins, carbohydrates, phospholipids among other compounds, including primary metabolites might be present. We modified in the text using crude water-soluble extract.
(iii) L124 - "Microcystins LR, RR, LA and YR were not detected in the strain NPCD-01, including the experiment by LC-HR‐QTOF‐MS as described above. Therefore, strain NPCD-01 was defined as a MC non-producer" This line is unjustified since there are many other microcystin congeners besides the four listed. Using an Adda ELISA or a less specific LC-MS technique such precursor ion scanning (https://www.mdpi.com/2072-6651/10/4/147/htm) would be a more believable way to identify the absence of MCs in the extract.
Authors’ Answer: We totally agree with referee 2. This sentence is unappropriated since we did not mention that microcystins were already analysed by ELISA in the strain NPCD-01. The strain NPCD-01 is part of CENA-USP cyanobacteria collection and a study performed by Laughinghouse IV et al (2012) [8] had already demonstrated by ELISA that NPCD-01 is a microcystin non-producer. The same strain was analysed by a direct competitive ELISA based on polyclonal antibodies (Beacon Analytical Systems Inc., USA) and microcystins were not detected in Microcystis aeruginosa NPCD-01. We added this information in the discussion section. Besides, previously, our group performed the analyses of these two Microcystis species by HPLC-DAD (first version of this manuscript), monitoring microcystin by its lmax (UV 234 nm) and results showed that NPCD-01 contains no microcystins.
In fact, precursor ion scan for the losses of m/z 135 from Adda and m/z 265 for ADMAdda containing compounds were not checked. However, recently, we had also looked for these two losses in our MS2 QTOF raw data and we did not find peaks containing the diagnostic losses of m/z 135 from Adda or m/z 265 for ADMAdda. We also added these information on the manuscript in the M&M and in the results sections (Figure S3 - Supplementary material).
REFERENCES
1. Turner, A. D.; Dhanji-Rapkova, M.; O’Neill, A.; Coates, L.; Lewis, A.; Lewis, K. Analysis of Microcystins in Cyanobacterial Blooms from Freshwater Bodies in England. Toxins (Basel). 2018, 10, doi:10.3390/toxins10010039.
2. Menezes, C.; Churro, C.; Dias, E. Risk Levels of Toxic Cyanobacteria in Portuguese Recreational Freshwaters. Toxins (Basel). 2017, 9, doi:10.3390/toxins9100327.
3. Kohoutek, J.; Adamovský, O.; Oravec, M.; Šimek, Z.; Palíková, M.; Kopp, R.; Bláha, L. LC-MS analyses of microcystins in fish tissues overestimate toxin levels—critical comparison with LC-MS/MS. Anal. Bioanal. Chem. 2010, 398, 1231–1237, doi:10.1007/s00216-010-3860-z.
4. Welker, M.; Maršálek, B.; Šejnohová, L.; von Döhren, H. Detection and identification of oligopeptides in Microcystis (cyanobacteria) colonies: Toward an understanding of metabolic diversity. Peptides 2006, 27, 2090–2103, doi:https://doi.org/10.1016/j.peptides.2006.03.014.
5. Ersmark, K.; Del Valle, J. R.; Hanessian, S. Chemistry and Biology of the Aeruginosin Family of Serine Protease Inhibitors. Angew. Chemie Int. Ed. 2008, 47, 1202–1223, doi:10.1002/anie.200605219.
6. Fewer, D. P.; Jokela, J.; Paukku, E.; Österholm, J.; Wahlsten, M.; Permi, P.; Aitio, O.; Rouhiainen, L.; Gomez-Saez, G. V; Sivonen, K. New Structural Variants of Aeruginosin Produced by the Toxic Bloom Forming Cyanobacterium Nodularia spumigena. PLoS One 2013, 8, 1–10, doi:10.1371/journal.pone.0073618.
7. Kisugi, T.; Okino, T. Micropeptins from the Freshwater Cyanobacterium Microcystis aeruginosa (NIES-100). J. Nat. Prod. 2009, 72, 777–781, doi:10.1021/np800631t.
8. Laughinghouse, H. D.; Prá, D.; Silva-Stenico, M. E.; Rieger, A.; Frescura, V. D. S.; Fiore, M. F.; Tedesco, S. B. Biomonitoring genotoxicity and cytotoxicity of Microcystis aeruginosa (Chroococcales, Cyanobacteria) using the Allium cepa test. Sci. Total Environ. 2012, 432, 180–188, doi:10.1016/j.scitotenv.2012.05.093.